# Genetic and Environmental Factors Influence the Pleomorphy of *LRRK2* Parkinsonism

**DOI:** 10.3390/ijms22031045

**Published:** 2021-01-21

**Authors:** Vinita G. Chittoor-Vinod, R. Jeremy Nichols, Birgitt Schüle

**Affiliations:** Department Pathology, Stanford University School of Medicine, Stanford, CA 94305, USA; chittoor@stanford.edu

**Keywords:** Parkinson’s disease, parkinsonism, *LRRK2*, neuropathology, modifier, genetics, GWAS, environmental risk factors, polygenic risk score

## Abstract

Missense mutations in the *LRRK2* gene were first identified as a pathogenic cause of Parkinson’s disease (PD) in 2004. Soon thereafter, a founder mutation in *LRRK2*, p.G2019S (rs34637584), was described, and it is now estimated that there are approximately 100,000 people worldwide carrying this risk variant. While the clinical presentation of *LRRK2* parkinsonism has been largely indistinguishable from sporadic PD, disease penetrance and age at onset can be quite variable. In addition, its neuropathological features span a wide range from nigrostriatal loss with Lewy body pathology, lack thereof, or atypical neuropathology, including a large proportion of cases with concomitant Alzheimer’s pathology, hailing *LRRK2* parkinsonism as the “Rosetta stone” of parkinsonian disorders, which provides clues to an understanding of the different neuropathological trajectories. These differences may result from interactions between the LRRK2 mutant protein and other proteins or environmental factors that modify LRRK2 function and, thereby, influence pathobiology. This review explores how potential genetic and biochemical modifiers of LRRK2 function may contribute to the onset and clinical presentation of *LRRK2* parkinsonism. We review which genetic modifiers of *LRRK2* influence clinical symptoms, age at onset, and penetrance, what *LRRK2* mutations are associated with pleomorphic *LRRK2* neuropathology, and which environmental modifiers can augment *LRRK2* mutant pathophysiology. Understanding how *LRRK2* function is influenced and modulated by other interactors and environmental factors—either increasing toxicity or providing resilience—will inform targeted therapeutic development in the years to come. This will allow the development of disease-modifying therapies for PD- and *LRRK2*-related neurodegeneration.

## 1. Introduction

In 2004, missense mutations in the *LRRK2* (leucine-rich repeat kinase 2) gene were identified in the Japanese Sagamihara kindred as a pathogenic cause of Parkinson’s disease (PD), as well as in families of other ethnic backgrounds across the world [1,2]. While the clinical presentation of motor symptoms of *LRRK2* parkinsonism has been largely indistinguishable from sporadic PD, there is a striking difference in the age at onset (AAO), disease penetrance, and neuropathological features, ranging from nigrostriatal loss with inconsistent occurrences of Lewy body (LB) and Alzheimer’s pathology, lack thereof, or atypical neuropathology. These differences in disease presentation could result from interactions between the LRRK2 mutant protein and certain protein or environmental modifiers. Its clinical and neuropathological heterogeneity implies that other factors could influence pathobiology, which might require or yield distinct therapeutic approaches. In this review, we describe (1) the clinical and neuropathological similarities and differences of *LRRK2* parkinsonism with sporadic or idiopathic PD (iPD); (2) what genetic modifiers of *LRRK2*, including Mendelian PD genes, GWAS risk factors, and polygenic risk scores, modify disease risk; (3) what *LRRK2* mutations are associated with pleomorphic *LRRK2* neuropathology; (4) what environmental modifiers affect *LRRK2* parkinsonism.

### 1.1. Clinical Presentation and Incidence of PD

PD is a common neurodegenerative movement disorder affecting dopaminergic neurons in the substantia nigra and other projection neurons in both the central and peripheral nervous systems [3,4]. The overall prevalence of PD in people over 45 years of age was estimated at 572 per 100,000 in 2010 in the US [5]. Approximately 930,000 people in the US were affected with PD in 2020, and, based on US Census Bureau population projections, it will rise to 1,238,000 by 2030 [5]. Clinically, distinct motor and nonmotor symptoms lead to a diagnosis of PD [6,7], although the rate of misdiagnosis is quite high, with only 58% accuracy for the initial diagnosis of PD [8]. Motor symptoms are comprised of bradykinesia, resting tremor, rigidity, and postural instability with an asymmetrical onset of symptoms. Nonmotor symptoms include loss of sense of smell, sleep/REM disorder, dysphagia, autonomic dysfunction such as constipation, urinary problems, changes in heart rate variability, and psychiatric problems with anxiety and depression as well as cognitive decline [9,10,11]. More men than women are diagnosed with PD. While the incidence ratio is <1.2 in males and females under the age of 50, it increases to 1.6 over 80 years of age [12]. About 10–15% of all PD patients and about 25% of early-onset PD cases in a clinical setting report a family history of PD [13].

### 1.2. LRRK2 Variants, Haplotypes, and Penetrance PD

The *LRRK2* p.G2019S (NM_198578.4 (*LRRK2*): c.6055G > A) variant is estimated to occur in approximately 1% of patients with sporadic PD and 3–6% of patients with familial PD in the United States; it shows the highest penetrance of all reported *LRRK2* risk variants (Table 1). The *LRRK2* p.G2019S mutation is common in the European, Middle Eastern, and North African populations. There is a wide range for AAO of mutation carriers, ranging from the third to the seventh decade of life [14]. Carrier frequency in Ashkenazi Jewish PD patients is 15–20%; for North African Berbers, it is up to 40% [15,16]. Several groups have studied the genomic background of the *LRRK2* p.G2019S haplotype, and data suggest that there have been at least three different founding events that gave rise to the *LRRK2* p.G2019S mutation between 1500–4500 years ago [17,18,19].

Penetrance for *LRRK2* p.G2019S has been reported to be variable, ranging from 24–100%, and penetrance seems to increase with age. This wide range can be attributed to differences in study cohorts, ethnic groups, the presence of genetic or environmental modifiers, and recruitment or analysis methods [20,21]. In the Michael J. Fox Foundation (MJFF) Ashkenazi consortium, studying 2270 relatives of 474 cases of Ashkenazi Jewish descent, a penetrance of 26% was found [20]. Comparison of penetrance for non-Ashkenazi Jewish carriers with Ashkenazi Jewish carriers did not differ significantly, with 25–42.5% at age 80 [22].

Besides the *LRRK2* p.G2019S mutation, many missense and some loss-of-function mutations have been described. However, proof of pathogenicity is pending on many of these mutations, owing to their rarity and scarce functional studies [19,23,24]. Importantly, loss of function mutations, such as stop-codon and frameshift mutations, were shown to not contribute to PD risk [24,25]. In a comprehensive meta-analysis, including 94 articles covering 49,299 cases and 47,319 controls, a number of common variants were assessed for the risk of developing PD. The *LRRK2* p.A419V (odds ratio (OR) 2.45), p.R1441C/G/H (OR 12.75), p.R1628P (OR 2.13), p.G2019S (OR 13.16), and p.G2385R (OR 2.27) mutations were associated with increased PD risk, whereas p.R1398H (OR 0.81) was associated with a decreased risk for PD [26] (Table 1).

### 1.3. LRRK2 Domain Structure and the Impact of Inherited Mutations

LRRK2 is a large, multidomain protein of 2527 amino acids (aa), with both GTPase and kinase enzymatic domains (Figure 1). The amino terminus (1–1287aa) is not essential for intrinsic kinase or GTPase activity [27] but participates in the regulation of LRRK2. The amino terminus contains armadillo and ankyrin repeats, the namesake leucine-rich repeat domain, and a cluster of crucial phosphorylation residues (Ser910 and Ser935 [28,29]). Full kinase catalytic activity requires the remainder of the protein (1326–2527aa [27,30]). This minimal catalytic fragment includes an active GTPase domain, termed Roc (Ras of complex proteins), which is juxtaposed to the COR (C-terminal of ROC) domain, classifying LRRK2 as a ROCO family protein [31,32]. The GTPase domain of LRRK2 can be purified as an active monomer/dimer that binds and hydrolyzes GTP [33], implicating that the RocCOR domain likely participates in the dimerization of LRRK2 [34,35,36]. The adjacent kinase domain bears similarity to mixed-lineage kinases, which are typically involved in kinase signaling cascades. The carboxy terminus contains a WD40 domain and is essential for kinase activity, where deletion of the last seven amino acids inactivate LRRK2 via disruption of the WD40 fold [27,37,38].

High-risk variants in the *LRRK2* gene encode substitutions in the catalytic core of the LRRK2 GTPase and kinase domains (p.N1437H, p.R1441C/G/H, p.Y1699C, p.G2019S, p.I2020T; Table 1 and Figure 1) [30,39,40,41,42]. The *LRRK2* p.G2019S mutation is located in subdomain VII of the kinase domain “Asp-Phe-Gly (DFG)” motif [43,44,45] and displays a 2- to 3-fold increase in kinase activity in vitro and in vivo [28,42,46,47]. Structural studies suggest that the serine substitution at position 2019 for the glycine in the DFG motif stabilizes an active state of LRRK2, resulting in the increased kinase activity [47,48,49]. The *LRRK2* p.I2020T mutation displays decreased activity in some assays and increased activity in others [27,30,50,51], possibly due to substrate-dependent readouts of kinase activity [52]. Mutations in the RocCor domain, p.N1437H, p.R1441C/G/H, and p.Y1699C, are thought to disrupt GTP hydrolysis, leaving LRRK2 in a GTP-bound, active kinase state, mediating increased kinase activity [33]. Noncatalytic domains of LRRK2 can also harbor PD, causing mutations. The *LRRK2* p.G2385R mutation decreases kinase activity of LRRK2 in vitro [27,37,38,53] but displays reduced Hsp90 interaction and 14-3-3 binding, with increased activity against Rab substrates in cells [30]. Conversely, *LRRK2* p.G2385R de-stabilizes LRRK2 dimers isolated from cultured cells and shows significantly elevated kinase activity [54]. Overall, there is a correlation between PD-causing LRRK2 mutations and an elevation in LRRK2 kinase activity or higher net signaling of LRRK2, which could mediate a differential response to upstream modifiers and provide a plausible explanation for the variability in their associated PD pathophysiology.

### 1.4. LRRK2 Parkinsonism: Similarities and Differences to iPD

*LRRK2* parkinsonism has been hailed as the “Rosetta stone” of parkinsonian disorders as brains from *LRRK2* cases demonstrate all major pathologies associated with parkinsonism, including a large proportion with concomitant Alzheimer’s pathology [55,56,57,58,59]. LRRK2 could hold the key to unlocking answers as to why different neuropathological changes can arise from underlying genetic changes in this gene. Currently, the pathways and potential disease modifier genes that explain this striking pleomorphic presentation of *LRRK2* parkinsonism and variable clinical penetrance/AAO are unknown, although many cellular phenotypes and molecular pathways, including endolysosomal stress [60,61,62], neuroinflammation [63,64,65], mitochondrial dysfunction and damage [66,67], and alterations of exosomes [68,69] have been described due to altered LRRK2 signaling. Mendelian PD genes, GWAS risk factors, and polygenic risk scores have been nominated as influencing factors for *LRRK2*, but studies that validate these putative modifiers are lacking.

#### 1.4.1. Motor and Nonmotor Features in *LRRK2*-PD

*LRRK2* parkinsonism can present with the same cardinal motor and nonmotor symptoms as sporadic PD [83] (we will refer to such patients and cohorts as *LRRK2*-PD), although the disease course and progression appear to be slightly more benign, as described in a recent meta-analysis of 66 clinical research studies evaluating the clinical phenotype of PD cases with p.G2019S, p.G2385R, p.R1628P, and p.R1441G *LRRK2* mutations [84]. Overall, *LRRK2*-PD presents with higher rates of early-onset PD, a higher female ratio, and family history, which can be explained by the strong genetic contribution [85]. PD patients with *LRRK2* p.G2019S have lower depression rates and higher daily activity scores as well as better olfactory function compared to sporadic PD [84]. *LRRK2* p.G2019S and p.G2385R have a good response to a higher daily dose of L-dopa, albeit with more motor complications than PD non-*LRRK2* carriers. *LRRK2* p.G2385R-affected carriers present with milder motor scores and better cognitive function. No distinct clinical features for *LRRK2* p.R1628P or p.R1441G have been detected [84]. In a prospective study of 144 *LRRK2* p.G2019S-PD and 401 non-*LRRK2*-PD, a slower decline in motor function was described among those with *LRRK2* p.G2019S-associated PD [86]. Interestingly, relatives of *LRRK2*-PD (*n* = 142, independent of mutation carrier status) present with a worse motor score and anxiety compared to 172 controls, implying that other environmental or non-*LRRK2* genetic modifiers might influence the penetrance of *LRRK2* p.G2019S-PD [87]. Nonmotor features and early predictors of PD also include reduced heart-rate variability (HRV) and cardiac sympathetic neurodegeneration [88,89]. In a study using classic HRV parameters, there was no difference between 20 *LRRK2*-PD and 32 healthy controls [90]. However, a follow-up study of an overlapping cohort analyzed novel HRV parameters and showed that there was more vagal involvement in 14 *LRRK2*-PD compared to 27 healthy controls and even in a subset of *LRRK2* non-manifesting carriers (a total group of 25 non-manifesting *LRRK2* carriers), suggesting a preclinical endophenotype of impairment of cardiac innervation [91].

#### 1.4.2. Cortico-Striato-Nigral Connectivity Alterations in Asymptomatic *LRRK2* p.G2019S Mutation Carriers

As disease penetrance for *LRRK2* p.G2019S is between 24–100% [20,21], there is interest in understanding the effect of this high-risk variant in asymptomatic carriers in order to develop disease-specific biomarkers and endophenotypes that could support early disease detection. Functional imaging studies of asymptomatic *LRRK2* p.G2019S carriers compared to healthy controls have shown changes in executive function and reward-based neural processing, suggesting alterations in neuronal networks and connectivity. Asymptomatic *LRRK2* p.G2019S mutation carriers (*n* = 27) show a reorganization of corticostriatal circuits similar to iPD, which is more pronounced with the increased age of *LRRK2* p.G2019S carriers compared to asymptomatic non-*LRRK2* carriers (*n* = 32) [92]. In resting-state magnetic resonance imaging (MRI), there was reduced integrity of nonmotor networks detected in asymptomatic *LRRK2* p.G2019S (*n* = 44) compared to asymptomatic non-*LRRK2* carriers (*n* = 41) before changes in the connectivity of the motor network were present, which illustrates that nonmotor cerebral changes delineate *LRRK2* p.G2019S carriers as “at risk” for developing PD [93]. Another study using resting-state MRI also found that altered brain connectivity precedes the onset of PD motor features. Asymptomatic *LRRK2* p.G2019S carriers (*n* = 18) showed functional connectivity changes in striatocortical and nigrocortical circuits compared to non-*LRRK2* first-degree relatives (*n* = 18) [94]. The connectivity alterations in asymptomatic *LRRK2* carriers may represent neural compensatory mechanisms. In the Stroop color-word interference test, *LRRK2* p.G2019S carriers (*n* = 19) had similar behavioral performance compared to 21 first-degree relative asymptomatic non-*LRRK2* carriers but had increased activity in brain regions comprising the ventral attention system [95]. In a study testing an event-related functional MRI gambling task to assess the reward network, abnormal neural activity in the reward and motor networks were detected in asymptomatic *LRRK2* p.G2019S carriers (*n* = 36) compared to controls (*n* = 32), indicating the involvement of the ventral striatum [96]. Another example of potential compensatory functional changes is that manifesting (*n* = 14) and non-manifesting (*n* = 16) *LRRK2* carriers exhibit increased cortical cholinergic activity compared to iPD (*n* = 8) and controls (*n* = 11) [97]. Overall, these findings show that circuit structure and function in *LRRK2* p.G2019S carriers are already altered before symptom onset and, presumably, already reorganized during neuronal development and brain maturation, suggesting that *LRRK2* p.G2019S carriers require compensatory mechanisms for normal cognitive function, which makes them potentially more susceptible to PD. While the studies include a small number of participants and are potentially underpowered, these are critical studies that further investigation. This invokes an interesting question of whether these compensatory functional effects are related to *LRRK2* modifiers and could lead to parallel pathways that also affect PD pathophysiology.

### 1.5. Pleomorphic LRRK2 Neuropathology

The neuropathological changes that underlie the clinical PD spectrum are subtyped as PD, dementia with Lewy bodies (DLB), and Parkinson’s disease dementia (PDD); they are characterized by the formation of intracellular protein inclusions and immunoreactive for alpha-synuclein (α-syn) and its pathological forms such as phosphorylated S129 α-syn [39,98]. The neurons most vulnerable for LB pathology are projection neurons with disproportionately long axons and poor myelination, exemplified by dopaminergic neurons projecting from the substantia nigra to the striatum [3,99]. In addition, it is becoming more evident that there is overlapping pathology in older adults, with up to 20% of quadruple neuropathology (tau neurofibrillary tangles, amyloid-β (Aβ), α-syn, and transactive response DNA-binding protein 43 (TDP-43)) in cases with dementia associated with a progressive course of disease [100].

After early reports of *LRRK2* neuropathology in human postmortem cases, it became apparent that *LRRK2* parkinsonism presents with heterogeneous neuropathology, including pure nigral–striatal degeneration or typical LB pathology, but also cases with multiple system atrophy (MSA) or progressive supranuclear palsy (PSP) staining with variable concomitant Alzheimer’s pathology [55,101,102]. This neuropathological heterogeneity cannot be explained by allelic variation in functional domains of LRRK2 since variability has been reported for pathology associated with all variants (Table 2, Figure 2). In iPD, approximately 5% of cases with clinically diagnosed PD present with postmortem neuropathological findings of pure nigrostriatal or substantia nigra degeneration (SND), while 77% are an LB disease, including brainstem predominant, transitional, and diffuse LB disease. MSA is found in 5% of clinical PD cases; PSP, especially the parkinsonism form (PSP-P), is found in 11% of probable PD according to Hoehn and Yahr staging without dementia [98]. Notably, these numbers are quite different in *LRRK2* parkinsonism cases (Figure 2). In *LRRK2* parkinsonism, the number of cases with pure substantia nigra degeneration is 33% (24 out of 73), while cases with typical LB pathology only comprise 38% (28 out of 73; Table 2). Mutations in the three catalytic core domains, Roc, COR, and kinase, can present with nigrostriatal degeneration, which includes p.R1441C/G/H, p.Y1699C, p.G2019S, and p.I2020T [55,56,57,103].

While rare, three pathogenic *LRRK2* cases with pathologically proven MSA have been described, which represent about 4% (3 out of 73 *LRRK2* cases) of all reported *LRRK2* parkinsonism cases. All cases harbored different mutations: one is located in the Roc domain (*LRRK2* p.I1371V [58]), and two are in the kinase domain (*LRRK2* p.I2020T [104] and *LRRK2* p.G2019S [105]). Histopathologically, in MSA, α-syn is found as aggregates, called glia cytoplasmic inclusions (GCIs), in oligodendroglia rather than neurons and are distinct from LBs. *LRRK2* has been found in oligodendroglia in MSA and is colocalized with GCIs as well as degrading myelin sheaths, which are one of the earliest cellular abnormalities that have been described in MSA [106]. One *LRRK2* p.G2019S case, which also involved the MAPT variant p.Q124E, was described with occasional TDP-43 inclusions, nigral degeneration without Lewy bodies, and Alzheimer-type tau pathology [107].

Approximately 22% of all *LRRK2* cases (16 out of 73) have been described with neuropathological changes of hyperphosphorylated tau resembling PSP (Figure 2, Table 2). PSP is a 4R tauopathy with tau aggregates in neurons and glia in specific neuroanatomical regions such as basal ganglia, diencephalon, brainstem, and cerebellum. One case with *LRRK2* p.R1441C, four with *LRRK2* p.I2020T, and five with *LRRK2* p.G2019S have been described in the literature with PSP [1,108,109,110,111]. Additionally, two cases with the risk variant *LRRK2* p.R1628P and one variant of unknown significance (*LRRK2* p.A1413T) have been reported with neuropathological changes compatible with PSP [112]. Two cases with *LRRK2* variants (p.R1707K and p.R2618P) have also shown neuropathology of corticobasal degeneration (CBD) [112].

Coexistent Alzheimer’s pathology is well known as part of the natural history of PD. Especially in PD cases with cognitive decline, 94% of them have been described with concomitant neuropathological changes of AD, whereas cases without dementia rarely present with AD pathology [125]. In *LRRK2* parkinsonism, a large proportion of cases with clinically diagnosed PD or PDD also have significant concomitant Alzheimer’s disease pathology with or without LBs (12 out of 15 cases) [56,110].

In conclusion, it appears that *LRRK2* parkinsonism cases with pure nigrostriatal degeneration are six times more frequent compared to iPD (33% vs. 5%), whereas typical LB pathology is reported at half the frequency as iPD (38% vs. 77%). MSA seems to occur at similar frequencies between *LRRK2* parkinsonism and sporadic PD, but *LRRK2* cases with PSP are 2.5 times higher than iPD (22% vs. 8%; Figure 2). These striking differences in neuropathological characteristics indicate that there are likely different disease modifiers of mutant LRRK2 or environmental modifiers that affect neurodegenerative trajectories of α-syn and tau pathology in *LRRK2* parkinsonism in vulnerable cell populations. A deficiency in the lysosomal–autophagy catabolic system is an attractive explanation for the causes of neurodegenerative diseases due to the overlap in phenotypes of several neurodegenerative model systems and the overall observation of a decrease in lysosomal functionality with age. However, a lysosomal centric explanation of cell death is insufficient to explain the increase in 4R tau for PSP and the increase in SND pathology in *LRRK2* parkinsonism. To try to understand how LRRK2 mutations contribute to the shift in the proportion of α-syn and tau neuropathologies, alternative explanations need to be found. For example, for the contribution to *LRRK2*-mediated PSP, a transgenic *LRRK2* mouse model showed an increase in 4R tau [126], which could indicate an effect of *LRRK2* on tau splicing. Lack of protein α-syn and tau aggregation in *LRRK2* parkinsonism leaves an open question, and other factors such as gene regulation, immune response, or mitochondrial dysfunction could be at play.

## 2. Environmental Modifiers of *LRRK2* Parkinsonism

### 2.1. Environmental and Lifestyle Factors Influence Penetrance and Age at Onset of LRRK2 Parkinsonism

Environmental and lifestyle factors, head injury, and exposure to industrial toxicants, including pesticides, solvents, or metals, among other pollutants, have been found to increase the risk for PD. Effects of these environmental exposures have been in animal models [127,128,129]. Some environmental factors have been associated with a decreased prevalence of PD, including smoking history, caffeine and tea consumption, use of nonsteroidal anti-inflammatory drugs (NSAIDs), and physical activity [130]. Lifestyle changes such as dietary alterations have been considered for their protective effects on neurodegenerative diseases, including AD and PD [131]. Such dietary changes can even be investigated in fly models with *LRRK2* p.G2019S, and they have shown a protective effect of dietary amino acids on dopamine neuron survival and motor function [132].

It is intriguing to hypothesize that environmental or lifestyle factors also influence the risk for *LRRK2*-PD, specifically disease penetrance and AAO. Interestingly, a study showed that *LRRK2* carriers with a history of cigarette smoking had a later AAO compared to nonsmokers, suggesting that smoking could modify AAO in *LRRK2*-PD [133]. Furthermore, regular use of NSAIDs (ibuprofen and aspirin) is associated with reduced penetrance in *LRRK2*-associated PD. This study included 259 *LRRK2*-PD and 318 *LRRK2*-asymptomatic participants; regular NSAID use resulted in reduced risk for PD in the overall cohort (OR: 0.34), including *LRRK2* p.G2019S, p.R1441C/G, p.I2020T, p.G2385R, and p.R1628P variants [134]. With consortium efforts, these emerging studies in *LRRK2* cohorts will hopefully be validated and expanded for other modifying environmental factors in the future.

### 2.2. Increased Susceptibility to Synthetic Toxicants in LRRK2 Animal Models

Environmental toxicants associated with the manifestation of PD mainly comprise inhibitors of mitochondrial complexes and/or inducers of cellular reactive oxygen species (ROS) [129]. These toxicants have been intensively investigated in animal models of PD. LRRK2 mutant protein contributes to multiple pathways and mechanisms, including neuroinflammation [64], endolysosomal and oxidative stress [61,135,136], and mitochondrial dysfunction and clearance [67,137]. Variable penetrance and AAO also suggest multiple hits for disease development [138,139], including predisposing germline gene mutations, acquired somatic gene variants [140], the environmental factors described in Section 2.1, and aging [141].

1-Methyl-4-phenyl-1,2,3,6-tetrahydropyridine (MPTP), a contaminant of the synthetic opiate meperidine, was identified as a neurotoxic compound that leads to an acute form of parkinsonism [142,143]. Neurodegeneration associated with *LRRK2* p.G2019S expression is augmented in mice after MPTP administration; however, mice expressing human WT *LRRK2* exhibit a similar response to MPTP as non-transgenic animals [144,145]. Subtoxic exposure to MPTP increases motor impairment, dopaminergic cell loss, and astrocyte activation, which can be partially reversed by pharmacological LRRK2 kinase inhibition [145]. On the other hand, mice that lack the LRRK2 kinase domain do not show any difference in MPTP sensitivity when compared to WT animals [146]. MPTP administration increases LRRK2 transcript levels together with other PD-related proteins such as Parkin, PINK1, MUL1, and USP30 in both non-transgenic rodents and rhesus monkeys [147,148] and, thereby, could further contribute to the neurotoxicity in *LRRK2* mutant models.

Paraquat (PQ; 1,1′-dimethyl-4-4′-bipyridinium) is an herbicide that bears a close structural resemblance to MPP+ that interferes with electron transport in the mitochondria to promote the formation of ROS. Studies show that PQ exposure also increases motor impairment in LRRK2-overexpressing animals. Both WT and mutant LRRK2 variants evoke a similar response to PQ in rodents [149,150]. However, knock-down of endogenous *LRRK2* leads to PQ resistance in mice and flies [149,151,152]).

Rotenone is a broad-spectrum organic pesticide that occurs naturally in several plants. Rotenone is an inhibitor of mitochondrial complex I and has been widely used for neurotoxic modeling in PD [153,154]. Mice with *LRRK2* p.R1441G knock-in show clear motor deficits upon rotenone exposure compared to WT LRRK2 mice treated with rotenone [155]. Studies from *Drosophila* also advocate that rotenone treatment exacerbates dopamine neuron loss and motor impairment in *LRRK2* transgenic animals [156,157], although different *LRRK2* variants may display variable susceptibilities towards this neurotoxin [157]. Furthermore, LRRK2 kinase inhibitors render resistance to rotenone toxicity, as seen in rats and human Wharton’s-jelly-derived mesenchymal stromal cells (hWj-MSCs) [158,159]. Reinhardt et al. showed that sensitivity to rotenone and 6-OHDA (6-hydroxy-dopamine) is higher in iPSC-derived midbrain dopamine neurons when a p.G2019S mutation is introduced into the *LRRK2* locus [160]. Together, there is compelling evidence that exposure to environmental neurotoxins can act as a second hit to *LRRK2* pathology, thus exaggerating the associated pathology.

Similar to other dominant genetic PD transgenic models and models of gene–environment interactions on nigrostriatal toxicity [161,162,163], LRRK2 mutations seem to sensitize and augment the underlying toxicant-induced phenotype. Likewise, expression of SNCA p.A53T also increases susceptibility to toxicants [164,165]. Conversely, overexpression of recessive PD gene products (modeling reversal of inherited mutation) is protective against toxicants [166]. 

## 3. *LRRK2* Risk and Protective Variants in Neurodegeneration and Genetic Modifiers of AAO and Penetrance in *LRRK2* Parkinsonism

### 3.1. LRRK2 Modifier Risk Variants

Risk variants for PD derived from GWAS or candidate gene studies have been tested in association studies in *LRRK2* cohorts. Such PD risk variants are located in *SNCA*, dynamin-3 (*DNM3*), cyclin-G-Associated Kinase (*GAK*), brain-derived neurotrophic factor (*BDNF*), microtubule-associated protein tau (*MAPT*), bone marrow stromal cell antigen 1 (*BST1*), Ras-related protein Rab-29 (*Rab29/PARK16*), and vesicle-associated membrane protein 4 (*VAMP4*) (Table 3). While mechanistic interactions with LRRK2 have been described in experimental studies for these risk genes [167,168], only *SNCA* and *MAPT* single nucleotide polymorphism (SNP) variants have been found to replicate in several studies (Table 3). Candidate gene studies of *LRRK2* modifiers for risk/penetrance and AAO have been described predominantly for p.G2019S, p.G2385R, p.R1628P, and other risk variants such as DNM3 (Table 3). Six studies investigated the *LRRK2* p.G2019S mutation [169,170,171,172,173,174], two studies included the *LRRK2* p.G2385R [175,176] variant, two studies combined the *LRRK2* p.G2385R and p.R1628P variants, which are the more common variants in the Asian population [177,178]. Three studies included *LRRK2* noncoding and coding risk variants rs1491942 (intronic), rs7133914 (p.R1398H), and rs10878226, noncoding 2 kb upstream of the *LRRK2* gene [179,180]. Of the six studies that tested *SNCA* noncoding variants, three studies found that *SNCA* rs356219 was associated with an earlier AAO for *LRRK2* p.G2019S, p.G2385R, and p.R1628P in the European and Han Chinese populations [169,173,178]. Additionally, for noncoding variants in the *MAPT* gene, three of six studies showed either an association with disease risk or AAO [170,176,181].

Most of these clinical genetic studies of *LRRK2* modifiers have evaluated candidate genes, and such studies have been challenged for their lack of replication due to small study cohorts of *LRRK2* carriers, different types of *LRRK2* alleles, and various ethnic populations. There are two pressing questions in the field: why is it that not all *LRRK2* variant carriers develop PD (reduced penetrance), and why can the AAO be quite variable even within families (variable age at onset). GWAS studies have nominated close to 100 genetic risk loci for PD [182,183,184,185,186,187]. With large consortium efforts, it is possible to study genetic *LRRK2* modifiers with adequate statistical power [188]. While single loci only contribute to a small effect on PD risk, combining individual loci allows for the assessment of cumulative risk represented as polygenic risk scores (PRSs) and can explain ~30% of the heritable risk for PD [184]. Using a PD PRS from 89 variants reported, a large cohort of heterozygous *LRRK2* p.G2019S mutation carriers (833 *LRRK2* p.G2019S carriers: 439 PD and 394 unaffected) was analyzed, and the PD PRS was associated with a higher penetrance of PD (OR: 1.34; *p* = 0.005). This association with PRS and penetrance was stronger in younger individuals under 55 years (OR: 1.95; *p* = 0.004) [189]. The first GWAS study of penetrance and age-at-onset of PD among *LRRK2* mutation carriers included 1879 *LRRK2* mutation carriers (853 from 294 families and 1026 singletons; 96% *LRRK2* p.G2019S carriers). A variant located in the intronic region of *CORO1C* on chromosome 12 (rs77395454; *p* = 2.5 × 10^−8^, beta = 1.27, SE = 0.23, risk allele: C) met genome-wide significance and may modify the penetrance of *LRRK2* mutations [188]. In addition, a PRS derived from publicly available PD GWAS statistics was a significant predictor of penetrance, but not age of onset, suggesting that common PD-associated variants collectively increase the penetrance of *LRRK2* mutations [188].

Two recent studies investigated the clinical presentation of digenic carriers of *LRRK2* p.G2019S and *GBA* mutations and found that patients carrying mutations in both genes have milder motor and nonmotor symptoms (specifically, better olfaction) compared to *GBA* mutation carriers; however, cognitive scores did not differ among the groups studied. This suggests that *LRRK2* p.G2019S has a potential protective role on PD motor and nonmotor phenotypes [190,191].

### 3.2. Protective LRRK2 Variants in PD and MSA, but not for Essential Tremor or AD

Clinical genetic studies of *LRRK2* as a PD risk gene also discovered variants and haplotypes that are associated with a decreased risk for developing PD. The best-studied variant is *LRRK2* p.R1398H, which has been shown, in conjunction with p.N551K, to be protective for developing PD (Figure 1, Table 1). This variant was first described in a multicenter study and showed that individuals carrying the *LRRK2* p.N551K and p.R1398H variants had a 20% reduced risk of developing PD [192]. These findings were replicated in several studies of Asian populations [193,194,195,196]. Additionally, a study of 177 cases with pathologically confirmed MSA and 768 controls revealed protective variants for MSA p.M2397T, p.G1624G, and p.N2081D [197]. However, while *LRRK2* p.N551K and p.R1398H variants seem to have a protective effect for PD, this protective effect could not be shown for essential tremor [198] or AD [199].

*LRRK2* p.R1398H affects GTPase activity, promoting axonal outgrowth and Wnt signaling activity in cell culture models; this suggests that these effects are critical for neuronal maintenance and function [73]. Induced pluripotent stem cell (iPSC) lines are being made from these protective variants, and further studies that elucidate protective mechanisms of these *LRRK2* variants will be critical to advancing the understanding of *LRRK2* variant function [200,201].

### 3.3. LRRK2 Variants in Other Neurodegenerative Diseases

*LRRK2* variants have also been tested in neurodegenerative diseases such as PSP, AD, dementia with Lewy bodies (DLB), MSA, and essential tremor; specific *LRRK2* variants have also been linked to inflammatory diseases such as Crohn’s disease and infectious diseases such as leprosy.

Only 1 out of 3 studies found *LRRK2* variants in PSP, which included p.G2019S, p.R1441C, p.R1628P, and a novel p.A1413T variant [112,202,203]. These studies suggest that PSP can present with pathogenic *LRRK2* variants but at a very low frequency. A recent study has suggested a new intronic variant in the SLC2A13 gene, localized upstream of the *LRRK2* gene, is associated with survival in PSP. The underlying mechanism of this association might be mediated through a long noncoding RNA-regulated effect on *LRRK2* gene expression [204,205].

Furthermore, in a cohort of 772 clinical DLB patients that included cases with confirmed MSA pathology, only two pathogenic *LRRK2* variants (p.G2019S and p.R1441C) were found [206]. Other studies have shown no association of *LRRK2* variants with MSA, assessing *LRRK2* p.G2019S, p.R1628P, or p.G2385R [207,208,209]. No significant risk of AD for *LRRK2* variants p.R1628P, p.G2385R, p.N551K, p.G2019S, and p.I2020T was detected in a meta-analysis including 13 studies [210]; additionally, no association for essential tremor (ET) risk and *LRRK2* risk variants (p.L1114L, p.I1122V, p.R1441C, p.Y1699C, p.I2012T, p.G2019S, p.I2020T, and p.G2385R) has been found [211,212,213,214,215]. However, the only *LRRK2* variant that has been associated with ET is *LRRK2* p.R1628P. This study included 1277 subjects comprising 450 ET cases and 827 controls, which showed a 2-fold increased risk for ET (OR: 2.20, *p* = 0.0035) [216].

Patients with inflammatory bowel disease (IBD) have a higher risk of developing PD [217]. A meta-analysis concluded that patients with Crohn’s disease (CD) have a 28% increased risk of PD, and, similarly, ulcerative colitis patients have a 30% increased risk of PD compared to controls [218]. Gene association studies have found a genetic link between IBD and PD through *LRRK2* as a common risk gene for both diseases. Similar to PD, *LRRK2* p.N551K and p.R1398H variants are protective, whereas the *LRRK2* p.N2081D variant increases the risk for IBD (OR 1.6, *p* = 2.1 × 10^−6^) [71].

## 4. Summary and Perspective

In the 15 years since the discovery of *LRRK2* mutations as a cause for PD, clinical and basic research has made enormous progress toward understanding the role of *LRRK2* in PD. The work to date has been instrumental in the quest to develop PD-preventing or PD-modifying drugs, with LRRK2 as a target. However, these efforts are not without potential downfalls, as complete inhibition of LRRK2 induces lung pathology in nonhuman primates that also resembles LRRK2-knockout phenotypes in rodents [219,220]. Additionally, in this review, we highlight the crucial distinctions of *LRRK2* parkinsonism from iPD with respect to pathology and clinical presentation. We propose that these differences arise from the differential effects of a variety of genetic or environmental factors.

In this context, deep longitudinal profiling allowed to discern four underlying “ageotypes” or personal aging markers (liver, kidney, immunity, and metabolic dysfunctions) [221]; the balance between stress resistance or regeneration potential and environment (lifestyle, infection, microbiome, age) might be altered by *LRRK2* variants in one or more groups. Such a systems approach might be necessary to gain a full understanding of the mechanisms of *LRRK2* parkinsonism.

Does this present a treatment obstacle? If we approach the problem with the foreground hypothesis that differences in *LRRK2* parkinsonism from iPD are the result of how different “modifiers” exert their action on LRRK2 function, then we will gain the opportunity to not only understand how asymptomatic *LRRK2* carriers phenoconvert to PD but also to identify novel therapeutic targets and pathways that might even be relevant and translatable for iPD.

## Figures and Tables

**Figure 1 ijms-22-01045-f001:**
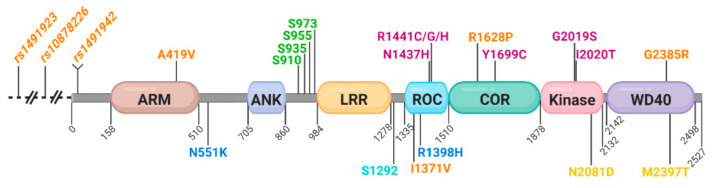
Domain architecture, genetic risk variants, and regulatory phosphorylation sites of LRRK2 (leucine-rich repeat kinase 2). LRRK2 is a large kinase of 2527 amino acids with multiple identifiable functional domains. Variants classified as pathogenic according to MDSGene are in magenta, while risk factor variants pending conclusive classification are in orange; putative protective mutations are in blue; variants associated with inflammatory bowel disease (IBD) are in yellow. Regulatory phosphorylation sites are in green, and *LRRK2* autophosphorylation site Ser1292 is in teal. ARM—armadillo domain; ANK—ankyrin domain; LRR—leucine-rich repeat domain; Roc—GTPase domain; COR—carboxy terminal of Roc domain.

**Figure 2 ijms-22-01045-f002:**
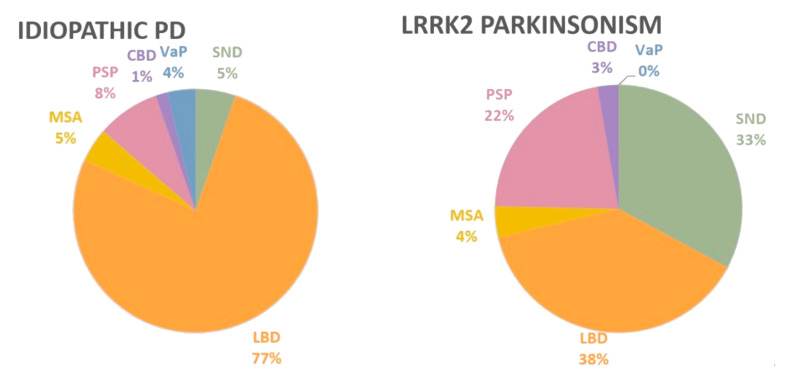
Frequency of neuropathologic disorders with a clinical diagnosis of idiopathic Parkinson’s disease (PD) and *LRRK2* parkinsonism. The pie charts show the frequency of pathologic disorders that present with clinical parkinsonism without dementia (*n* = 132), adapted from Dickson et al. (2018) [98], and *LRRK2*-PD from cases and case series described in the literature (*n* = 74; Table 1). Abbreviations: Lewy body disorder (LBD), multiple system atrophy (MSA), progressive supranuclear palsy (PSP), corticobasal degeneration (CBD), vascular parkinsonism (VaP), and substantia nigra degeneration (SND). For details on neuropathological assessments, see Table 2.

**Table 1 ijms-22-01045-t001:** *LRRK2* variant overview.

SNP (rs ID)	cDNA NM_198578.4	Protein XP_5268686.1	Odds Ratio (95% Confidence Interval)	Allele Frequency	Pathogenicity Score	Domain	Enzymatic Impact/Function
rs34594498	c.1256C > T	p.A419V	2.45 (1.43, 4.2) East Asian	0.0004854	Not reported	ARM	Does not alter kinase activity [70]
rs7308720	c.1653C > A, C > G	p.N551K	Not reported	0.000003985	Not reported	No domain	Does not alter kinase activity [70,71]; CD protection [71]
rs17466213	c.4111A > G	p.I1371V	Not reported	0.0008429	Probable	ROC	Increased GTP binding [72]
rs7133914	c.4193G > A	p.R1398H	0.81 (0.75, 0.89) Mixed	0.00001772	Not reported	ROC	Increased GTP binding and GTPase activity [73]; CD protection [71]
rs74163686	c.4309A > C	p.N1437H	Low frequency, not reported	Not found	Definite	ROC	Disrupts GTP hydrolysis, increased kinase activity [42,74]
rs33939927	c.4321C > T/A/G	p.R1441C/G/H	12.75 (3.11, 52.27)	0.00001195	Definite	ROC	Disrupts GTP hydrolysis; increased kinase activity [42,75,76]
rs33949390	c.4883G > T	p.R1628P	2.13 (Asian)	0.0001491	Not reported	COR	Increased kinase activity [77]; protective for T1R [78]
rs35801418	c.5096A > G	p.Y1699C	Low frequency, not reported	Not found	Definite	COR	Disrupts GTP hydrolysis; increased kinase activity [42,79]
rs34637584	c.6055G > A	p.G2019S	13.16 (10.16, 17.04) Mixed ethnicities	0.0004884	Definite	Kinase	Increased kinase activity [28,42,47,75]
rs35870237	c.6059T > C	p.I2020T	Low frequency, not reported	Not found	Definite	Kinase	Increased kinase activity in cells [42,50,72]
rs33995883	c.6241A > G	p.N2081D	Not reported	0.01685	Not reported	Kinase	CD risk and increased kinase activity [71]
rs34778348	c.7153G > A	p.G2385R	2.27 (2.03, 2.53) East Asian	0.001680	Possible	WD40	Increased kinase activity in cells [30,54]
rs3761863	c.7190T > C	p.M2397T	Not reported	0.6191	Not reported	WD40	CD risk enhances IFN-γ response [80]; T1R proinflammatory [81]; destabilizes protein [82]

Meta-analysis for odds ratio from Shu et al. 2019. DOI: 10.3389/fnagi.2019.00013. Frequency is based on gnomAD v2.1.1 (https://gnomad.broadinstitute.org/). Pathogenicity of variants was determined by MDSGene (https://www.mdsgene.org/).

**Table 2 ijms-22-01045-t002:** *LRRK2* cases with a neuropathological assessment.

*LRRK2* Mutation	Clinical Presentation	# of Cases	SND	LBD	MSA	PSP	CBD	TDP-43	Low AD	Interim AD	High AD	References
c.4111A > G (p.I1371V)	PD	1	-	1	-	-	-	-	-	-	-	[55,113]
	MSA	1	-	-	1	-	-	-	-	-	-	[58]
c.4309C > A (p.N1437H)	PD	1	-	1	-	-	-	-	1	-	-	[55,114]
c.4321C > T/G (p.R1441C/G)	PD	6	3	2	-	1	-	-	-	-	-	[115]
	PD	4	1	2	-	1	-	-	-	-	-	[1,116] (Family D, R1441C)
	PD	6	3	2	-	1	-	-	-	-	-	[1,55,115]
c.4322 G > A (p.R1441H)	PD	3	3	-	-	-	-	-	-	-	-	[57]
c.5096A > G (p.Y1699C)	PD	3	2	1	-	-	-	-	-	-	1	[55,117] (Lincolnshire, III.13), [1,118] (Fam A, III.27, III.29)
c.6055G > A (p.G2019S)	2 PD, 3 PDD	5	1	2	-	2	-	-	1	3	1	[24]
	7 PD, 2 PDD	9	4	5	-	-	-	-	-	-	3	[56,119] (3 cases)
	MSA	1	-	-	1	-	-	-	-	-	-	[105]
	PSP	-	-	-	-	2	-	-	-	-	-	[108]
	PD	1	1	-	-	-	-	-	1	-	-	[103]
	PSP	1	-	-	-	1	-	-	1	-	-	[112]
	PD	18	5	9	-	1	-	-	-	-	3	[55,120,121], 3 cases excluded from [119]
	PD	1	-	-	-	-	-	1	-	-	1	[107], patient also carries MAPT c.370C > G, (p.Q124E) variant
c.6059T > C (p.I2020T)	PD/MSA/PSP	9	5	1	1	4	-	-	-	-	-	Sagamihara [55,104,109,122,123]; (4 cases with tau pathology)
c.3494T > C, p.L1165P	PDD, Pat. E	1	-	1	-	-	-	-	-	-	1	[56,124]
c.2378 G > T, p.R793M	PD, Pat. D	1	-	1	-	-	-	-	-	1	-	[56,124]
c.5120G > A, p.R1707K	CBD	1	-	-	-	-	1	-	-	-	1	[112]
c.4883G > T, p.R1628P	PSP/CBD	3	-	-	-	2	1	-	-	-	-	[112]
c.4237G > A, p.A1413T	PSP	1	-	-	-	1	-	-	-	-	-	[112]
Total cases		74	24	28	3	16	2	1	4	4	9	

Lewy body disorder (LBD); multiple system atrophy (MSA); progressive supranuclear palsy (PSP); corticobasal degeneration (CBD); substantia nigra degeneration (SND); low, medium, high AD (levels of AD pathology).

**Table 3 ijms-22-01045-t003:** Genetic modifiers of penetrance and age at onset in *LRRK2* parkinsonism.

			Association with PD Risk Alleles									
*LRRK2* Variants	Sample Size	Population	SNCA	DNM3	GAK	BDNF	MAPT	BST1	Rab29/Rab7L1	VAMP4	PD-PRS	Ref.
rs1491942 (intronic)	1381 PD, 1328 ctrl	North America, Irish, Polish	rs356219, no	N/A	rs6599388 no	N/A	rs2942168 no	rs11724635 no	rs708723 no	N/A	N/A	[179]
rs1491942, rs7133914 (R1398H)	not reported	Caucasian, Asian (GEOPD)	N/A	N/A	N/A	N/A	N/A	N/A	No	N/A	N/A	[180]
p.R1441C, p.Y1699C, p.G2019S, p.I2020T	44 carriers, 19 families	European, North American	No	N/A	N/A	N/A	rs2435207, AAO	N/A	N/A	N/A	N/A	[181]
p.G2385R, p.R1628P	231 PD G2385R, 65 PD R1628P	Chinese	No	N/A	N/A	N/A	No	No	No	N/A	N/A	[177]
p.G2385R	64 PD	Chinese	N/A	N/A	N/A	AAO	N/A	N/A	N/A	N/A	N/A	[175]
p.G2385R	53 PD	Chinese	N/A	N/A	N/A	N/A	IVS1 + 124C > G increased risk (major allele)	N/A	N/A	N/A	N/A	[176]
p.G2385R, p.R1628P	82 PD G2385R, 46 PD R1628P	Han Chinese	rs356219, risk and AAO (OR: 1.5)	rs2421947 No	rs1564282 No	N/A	N/A	N/A	N/A	N/A	N/A	[178]
p.G2019S	84 PD	European	rs356219, AAO	N/A	N/A	N/A	N/A	N/A	N/A	N/A	N/A	[169]
p.G2019S	84 PD	Ashkenazi	N/A	N/A	N/A	N/A	rs11079727, AAO (older in minor allele)	N/A	N/A	N/A	N/A	[170]
p.G2019S	101 PD	Arab-Berber	No	N/A	N/A	N/A	No	N/A	N/A	N/A	N/A	[171]
p.G2019S	41 families: 150 PD, 103 unaffected, 232 unrelated	Arab-Berber	N/A	rs2421947, AAO earlier with GG allele	N/A	N/A	N/A	N/A	N/A	N/A	N/A	[172]
p.G2019S	210 PD, 119 unaffected	European (Spain)	rs356219, AAO	No	N/A	N/A	N/A	N/A	N/A	N/A	N/A	[173]
p.G2019S, rs10878226 (2 kb upstream)	724 PD p.G2019S, 4882 PD rs10878226	IPDGC and other	N/A	No AAO	N/A	N/A	N/A	N/A	N/A	No	N/A	[174]
p.G2019S	841 (439 PD, 394 unaffected)	European, North America	N/A	N/A	N/A	N/A	N/A	N/A	N/A	N/A	P (OR: 1.34)	[189]
	**# of Significant Studies (# Total Studies)**		**3 (6)**	**1 (4)**	**0 (2)**	**1 (1)**	**3 (6)**	**0 (2)**	**0 (3)**	**0 (1)**	**1 (1)**	

SNPs included per candidate gene study: SNCA: rs11931074, rs1372525, rs181489, rs2583988, Rep-1, and rs356219; DNM3: rs2421947, GAK: rs1564282; BDNF: p.V66M; MAPT: rs1052553, rs242562, rs2435207, H1/H2, IVS1 + 124C > G, rs393152, rs2435207, and rs11079727; BST1: rs4273468; Rab29/Rab7L1: rs823144, rs823139, rs708725, rs823156, rs11240572, and rs708723; VAMP4: rs11578699.

## Data Availability

This review used data available in publicly accessible repositories.

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
