# Peer review of "Genetic and Environmental Factors Influence the Pleomorphy of LRRK2 Parkinsonism"

_ijms, 2021, doi:10.3390/ijms22031045_

Round 1

Reviewer 1 Report

This is a timely review on genetic and environmental factors that influence LRRK2. It is clear, concise and focuses primarily on the genetic aspects, as the environmental factors are more challenging to measure (see additional comments below). The tables are comprehensive and helpful – distilling key information from the literature. Figure 2 is also helpful in representing the neuropathological differences between iPD and LRRK2 PD.

Major points:

  1. Section 1.2 – how is proof of pathogenicity defined? The text suggests that this is only clear for G2019S. Yet the Zimprich et al 2004 paper (PMID: 15541309) showed Y1699C and R1441C as familial mutations. Can the authors please elaborate on this?
  2. Table 1 lists G2019 as having the highest penetrance however some publications (e.g. PMID: 18337586 and 22315721) report higher penetrance for R1441C. Can the authors please elaborate on this?
  3. Table 1 – the authors should use frequencies from Gnomad https://gnomad.broadinstitute.org/ instead of the smaller ExAC data set.
  4. Section 1.3 (lines 110-113) – the reported effects of the G2385R mutation are mixed. However, the most recent research does suggest that there is an increase in kinase activity (see, and include, PMID: 30670570).
  5. Section 1.4 (and abstract) – reference to LRRK2 as the ‘Rosetta Stone’ is misleading given the varied pathology observed (nicely described in 1.5). The authors should remove this from the text.
  6. Sections 1.4.1 and 1.4.2 – it would be helpful to include the number of subjects evaluated in the studies that are referenced. Particularly in 1.4.2 as these studies are likely to be underpowered.
  7. Section 1.5 - Figure 2 + Table 2 are very interesting with regard to the different brain pathology of LRRK2-PD in terms of LB pathology (previously reported to be less prominent in LRRK2-PD) versus pure nigrostriatal degeneration (appears to be much higher in LRRK2-PD (33%) than iPD (~7%; or 5%?? -compare lines 231 with 197)). Is there any particular mechanism associated with the pure nigrostriatal cell death in LRRK2-PD that the authors could propose?
  8. Section 2. This is a tricky section due to the challenges of convincingly linking environmental factors to disease. Referencing the epidemiological (human) studies is fine (with all the caveats of these approaches and analyses). The authors have summarized the preclinical studies. However, the diversity of environmental toxins, model systems investigated, and endpoints used makes comparisons difficult. For example, this reviewer is not sure how to compare an MPTP study in mice over-expressing LRRK2 to human iPSC derived dopaminergic neurons exposed to 6-OHDA. This is not a direct critique of the authors per se, but more a point to the field in general that work is published, more often than not, with limited mechanistic understandings. If the authors want to include this section then they should try and provide more comprehensive links to the underlying mechanisms – e.g. neuroinflammation (PMIDs: 33055242; 32508566; 32410948), lysosomal stress (PMID: 33177079; 32919031; 32643832; 31804465) and further emphasize the multiple-hit hypothesis (e.g. PMID: 31192157).
  9. Table 2 – there is a column for TDP-43 pathology, but nothing in the it. Is this an error (c.f. PMID: 23664753)?
  10. Section 3.1 – there is no mention of GBA1 as a modifier (or more accurately of LRRK2 as a modifier of GBA1), c.f. PMIDs: 30573413; 32353202. This should be included.
  11. Section 3.3 – should include the recent finding that genetic variation at the LRRK2 locus was associated with survival in PSP (PMID: 33341150)

Minor points:

Line 62: NM_198578.3 can be updated to NM_198578.4

Line 319: Replace ‘do not’ with ‘don’t’

Line 380: Include this additional reference: PMID: 32321864

Line 385: Typo: ‘asymptimatic’

Author Response

Response to review ijms-1066162 with minor modifications: Reviewer 1. Comments and Suggestions for Authors. This is a timely review on genetic and environmental factors that influence LRRK2. It is clear, concise and focuses primarily on the genetic aspects, as the environmental factors are more challenging to measure (see additional comments below). The tables are comprehensive and helpful – distilling key information from the literature. Figure 2 is also helpful in representing the neuropathological differences between iPD and LRRK2 PD. Reply: Thanks so much for the positive, critical, and detailed review of our manuscript. The comments, additional references have really elevated the manuscript. We answer every point and update the manuscript accordingly. Major points: 1. Section 1.2 – how is proof of pathogenicity defined? The text suggests that this is only clear for G2019S. Yet the Zimprich et al 2004 paper (PMID: 15541309) showed Y1699C and R1441C as familial mutations. Can the authors please elaborate on this? Reply: The categorization of LRRK2 variants into causative and benign is an important and critical (sometimes controversial) point. It is not trivial for some mutations to define pathogenicity for various reasons: rarity of mutation, hence lack of population data, lack of functional data or models, and other. To be consistent for this review, we decided to follow the MDS Gene criteria (https://www.mdsgene.org/) as a growing database with expert review on PD gene variants. For mutations that are not included in the MDS Gene database, we use and cite source articles to reference. We have also included additional references for functional outcomes of the describes variants. 2. Table 1 lists G2019 as having the highest penetrance however some publications (e.g. PMID: 18337586 and 22315721) report higher penetrance for R1441C. Can the authors please elaborate on this? Reply: In Table 1, we refer to a large meta-analysis of 94 articles including 49,299 cases and 47,319 controls from different ethnic groups. It is possible that this is the explanation for the discrepancy of penetrance. However, both G2019S and R1441C are pathogenic variants with a high Odds ratio. 3. Table 1 – the authors should use frequencies from Gnomad https://gnomad.broadinstitute.org/ instead of the smaller ExAC data set. Reply: Thank you for the comment. We updated the frequencies based on the gnomAD dataset v2.1.1 and referenced link in the Table caption. 4. Section 1.3 (lines 110-113) – the reported effects of the G2385R mutation are mixed. However, the most recent research does suggest that there is an increase in kinase activity (see, and include, PMID: 30670570). Reply: We included the new reference and modified the text, section 1.3, lines 115-116: “Conversely, LRRK2 p.G2385R de-stabilizes LRRK2 dimers isolated from cultured cells and show significantly elevated kinase activity.” 5. Section 1.4 (and abstract) – reference to LRRK2 as the ‘Rosetta Stone’ is misleading given the varied pathology observed (nicely described in 1.5). The authors should remove this from the text. Reply: We use Rosetta stone with the meaning that “one that gives clue to understanding”. We think that LRRK2 will provide answers how different neuropathologies are linked together and develop. We modified the text lines and added our definition and context, abstract lines 15-17: “…hailing LRRK2 parkinsonism as the "Rosetta stone" of parkinsonian disorders which provides clues to an understanding of the different neuropathological trajectories.” And section 1.4, lines 123-125: “LRRK2 could hold the key to unlock answers as to why different neuropathological changes can arise from underlying genetic changes in this gene.” 6. Sections 1.4.1 and 1.4.2 – it would be helpful to include the number of subjects evaluated in the studies that are referenced. Particularly in 1.4.2 as these studies are likely to be underpowered. Reply: We updated both sections and included numbers of subjects per groups evaluated. We agree, that power is not always sufficient, nevertheless, we believe that these studies are important towards the understanding of LRRK2 mechanisms in the context of this review. 7. Section 1.5 - Figure 2 + Table 2 are very interesting with regard to the different brain pathology of LRRK2-PD in terms of LB pathology (previously reported to be less prominent in LRRK2-PD) versus pure nigrostriatal degeneration (appears to be much higher in LRRK2-PD (33%) than iPD (~7%; or 5%?? -compare lines 231 with 197)). Is there any particular mechanism associated with the pure nigrostriatal cell death in LRRK2-PD that the authors could propose? Reply: Thanks so much for the comment. We corrected the % to 5%, not 7%. We added some speculate thoughts re PSP and SND pathology. It is indeed unclear. We included the following text in section 1.5, lines 253-265:“A deficiency in the lysosomal-autophagy catabolic system is an attractive explanation for the causes of neurodegenerative diseases due to the overlap in phenotypes of several neurodegenerative model systems and overall observation of the decrease in lysosomal functionality with age. However, a lysosomal centric explanation of cell death is insufficient to explain the increase in 4R tau for PSP and the increase in SND pathology in LRRK2 parkinsonism. To try to understand how LRRK2 mutations contribute to the shift in the proportion of α-syn and tau neuropathologies, alternative explanations need to be found. For example for the contribution to LRRK2-mediated PSP, a transgenic LRRK2 mouse model shows an increase in 4R tau, which could indicate an effect of LRRK2 on tau splicing. Lack of protein α-syn and tau aggregation in LRRK2 parkinsonism leaves an open question and other factors such as gene regulation, immune response, or mitochondrial dysfunction could be at play.” 8. Section 2. This is a tricky section due to the challenges of convincingly linking environmental factors to disease. Referencing the epidemiological (human) studies is fine (with all the caveats of these approaches and analyses). The authors have summarized the preclinical studies. However, the diversity of environmental toxins, model systems investigated, and endpoints used makes comparisons difficult. For example, this reviewer is not sure how to compare an MPTP study in mice over-expressing LRRK2 to human iPSC derived dopaminergic neurons exposed to 6-OHDA. This is not a direct critique of the authors per se, but more a point to the field in general that work is published, more often than not, with limited mechanistic understandings. If the authors want to include this section then they should try and provide more comprehensive links to the underlying mechanisms – e.g. neuroinflammation (PMIDs: 33055242; 32508566; 32410948), lysosomal stress (PMID: 33177079; 32919031; 32643832; 31804465) and further emphasize the multiple-hit hypothesis (e.g. PMID: 31192157). Reply: That so much for the suggestion. We agree that gene-environment studies in transgenic models can be limited in scope and lack insightful mechanistic understanding, however, we think that the complex etiology of PD requires establishment of such models. We included the suggested references, which provides now additional context to the paragraph. We included the following text in section 2.2, lines 289-294: “LRRK2 mutant protein contributes to multiple pathways and mechanisms including neuroinflammation, endolysosomal and oxidative stress, or mitochondrial dysfunction and clearance. The variable penetrance and AAO also suggests multiple hits for disease development, including predisposing germline gene mutations, acquired somatic gene variants, environmental factors as described above in section 2.1, and aging.” 9. Table 2 – there is a column for TDP-43 pathology, but nothing in the it. Is this an error (c.f. PMID: 23664753)? Reply: Thanks so much for the comment, we included the suggested reference in Table 2. This case is a digenic carrier for LRRK2 and MAPT variants and we had also included in text, lines 228 -230: “One LRRK2 G2019S PD case, who also carried a MAPT variant p.Q124E, has been described with occasional TDP-43 inclusions, nigral degeneration without Lewy bodies and Alzheimer-type tau pathology.” 10. Section 3.1 – there is no mention of GBA1 as a modifier (or more accurately of LRRK2 as a modifier of GBA1), c.f. PMIDs: 30573413; 32353202. This should be included. Reply: These are indeed important new findings for digenic carriers of LRRK2 and GBA mutations and we include in section 3.1, lines 376-380: “Two recent studies investigated the clinical presentation of digenic carriers of LRRK2 p.G2019S and GBA mutations and found that patients carrying mutations in both genes have milder motor and non-motor symptoms (specifically better olfaction) as compared to GBA mutation carriers, however cognitive scores did not differ among the groups studied. This suggests that LRRK2 p.G2019S has a potential protective modifying role on the PD motor and non-motor phenotypes. “ 11. Section 3.3 – should include the recent finding that genetic variation at the LRRK2 locus was associated with survival in PSP (PMID: 33341150) Reply: Thanks so much for suggesting this manuscript. We included the reference and the commentary from Chen-Plotkin and added the following, section 3.3, lines 406-409: “A recent study, suggested a new intronic variant in the SLC2A13 gene, localized upstream of the LRRK2 gene, to be associated with survival in PSP. The underlying mechanism of this association might be mediated through a long non-coding RNA-regulated effect on LRRK2 gene expression.” Minor points: Line 62: NM_198578.3 can be updated to NM_198578.4 - updated Line 319: Replace ‘do not’ with ‘don’t’- changed Line 380: Include this additional reference: PMID: 32321864 – reference included Line 385: Typo: ‘asymptimatic’ – corrected.

Reviewer 2 Report

The LRRK2 manuscript presented by Chittoor-Vinod et al. presents relevant contributions, resulting from an exhaustive and selected review of the bibliography. It also presents contributions and conclusions derived from new interpretations of previously published data. Only some minor issues should be fixed before publication:

-Bibliography

The reference list used is too long (182), and sometimes does not correspond to the works cited in the text. For instance, it lacks reference 1 in the text. On the other hand, some statements are not based in any reference, as well as the data presented in tables (enzymatic impact/function). By the way, table 1 should appear before than figure 1, since is cited before in the text.

Some statements are based in 7 references (f.i. 133-139). In my opinion more than 4 is excessive.

The relation between missense mutations in LRRK2 and PD was firstly described in 2004 by Zimprich and Paisán-Ruiz simultaneously. The work cited as 2004 (japanese family), is actually published in 2002, and only links the locus to the disease (not the gene).

-Terminology

Sometimes use indistinctly PD and parkinsonism, and are different terms.

Sometimes one and three letter codes are used simultaneously to name mutations at the protein level (even in tables), and it is not necessary.

Mutations must be named according to the standard (f.i. p.G2019S or p. Gly2019Ser), and using always the same for. Most of mutations lack p.

-Contents

The mutations that appear in Figure 1 as causative are not the same as those previously defined (meta-analysis). Some of the mutations shown in this figure are not clearly causative (N1437H).

In table one is not described the meaning of some parametres (M-H, Fixed, ...) and some places are empty (without any explanation). Furthermore, the election of variants that appear in this table is not clear.

page numbers are repeated several times

line71-LRRK2 penetrance = 24-100% [20-21]. line154-LRRK2 penetrance = 25% (no reference)

Figure 2. SND and CBD are not cited in the text. Furthermore, the abbreviationa are not defined.

line 205.  R1441C/H are mentioned but not p.R1441G

Table 2.TDP-43 is not present in any mutation (therefore, leftover). Most of additional differences are not such or links (are institutions or Universities)

2.2 Increased susceptibility to synthetic oxidants in LRRK2 animal models

It is not clear if this phenotype is exclusive of LRRK2 mutations, or PD-related mutations in general

3.1 LRRK2 modifier risk variants

DNM3 also modifies AAO, but is not mentioned

l360-361 is in bold letters

ET (essential tremor) abbreviation is not defined in the text

Summary sections is seems a Discussion or Perspective section

Author Response

Reviewer 2.

Comments and Suggestions for Authors: The LRRK2 manuscript presented by Chittoor-Vinod et al. presents relevant contributions, resulting from an exhaustive and selected review of the bibliography. It also presents contributions and conclusions derived from new interpretations of previously published data. Only some minor issues should be fixed before publication:

Reply: Thanks so much for the positive, critical, and detailed review of our manuscript. The comments and suggestions have really elevated the manuscript. We replied to every point and updated the manuscript accordingly.

Reviewer comment 1: -Bibliography: The reference list used is too long (182), and sometimes does not correspond to the works cited in the text. For instance, it lacks reference 1 in the text. On the other hand, some statements are not based in any reference, as well as the data presented in tables (enzymatic impact/function). Some statements are based in 7 references (f.i. 133-139). In my opinion more than 4 is excessive.

Reply 1: We try to balance the number of references and updated the manuscript. Since there is no limit for references and one of the IJMS encouragements is to be comprehensive, we did not want to omit any references and recognize authors for their contributions to the field.

However, we carefully reviewed the references against the text and updated where necessary. We reduced the number of references per statement to 4 or less. In addition, we included additional references suggested by the other reviewer.

Reference 1. This was a formatting error. Reference 1 was listed in a figure legend for Figure 2. We corrected this mistake.

Table 1 – we added references for each variant for enzymatic impact/function.

Reviewer comment 2: By the way, table 1 should appear before than figure 1, since is cited before in the text.

Reply 2: Thanks so much, we moved Figure 1 below Table 1.

Reviewer comment 3: The relation between missense mutations in LRRK2 and PD was firstly described in 2004 by Zimprich and Paisán-Ruiz simultaneously. The work cited as 2004 (japanese family), is actually published in 2002, and only links the locus to the disease (not the gene).

Reply 3: We corrected the references and cited only Zimprich and Paisán-Ruiz. We cited the 2002 publication (with link to family and PARK8 locus) later in Table 2.

Reviewer comment 4: -Terminology: Sometimes use indistinctly PD and parkinsonism, and are different terms.

Reply 4: That is a good point. In principle, we adhere to the MDS Clinical Diagnostic Criteria for PD (Postuma  et al. 2015, 2018). In publications prior to 2015, we also accept UK brainbank criteria or Hoehn & Yahr classification. Where PD is specifically defined in the publications, we use the term “LRRK2-PD”. When we discuss clinical symptoms of LRRK2 variants in general, we use “LRRK2 parkinsonism” to also include atypical PD syndromes such as PSP, MSA, and CBD. Discrepancies have been updated and corrected throughout the manuscript.

Reviewer comment 5: Sometimes one and three letter codes are used simultaneously to name mutations at the protein level (even in tables), and it is not necessary. Mutations must be named according to the standard (f.i. p.G2019S or p. Gly2019Ser), and using always the same for. Most of mutations lack p.”

Reply 5: Thanks so much for the comment. We updated the naming convention throughout the manuscript and use 1-letter code and p. = protein, e.g. p.G2019S. In the Table 1, deleted the column with 3-letter code as it is redundant.

Reviewer comment 6: -Contents: The mutations that appear in Figure 1 as causative are not the same as those previously defined (meta-analysis). Some of the mutations shown in this figure are not clearly causative (N1437H).

Reply 6: The categorization of LRRK2 variants into causative and benign is an important and critical point (and can be controversial for some variants). It is not trivial for some mutations to clearly define them at this point for various reasons: rarity of mutation, hence lack of population data, and lack of functional data or models.

We decided to follow the MDS Gene criteria (https://www.mdsgene.org/) as a growing database with expert review on PD gene variants. For mutations that are not included in the MDS Gene database, we use and cite other source articles.

Reviewer comment 7: In table one is not described the meaning of some parametres (M-H, Fixed, ...) and some places are empty (without any explanation). Furthermore, the election of variants that appear in this table is not clear.

Reply 7: Thanks so much for the careful review of Table 1. We included all LRRK2 variants in this table that we described throughout the manuscript. We updated the column for Odds Ratio and indicate where information is missing as “not found” or “not reported” to avoid empty fields and provide explanation.

Reviewer comment 8: page numbers are repeated several times

Reply 8: Page numbers will be corrected during typesetting. Template does not correctly populate page numbers when format is changed from portrait to horizontal view.

Reviewer comment 9: line71-LRRK2 penetrance = 24-100% [20-21]. line154-LRRK2 penetrance = 25% (no reference)

Reply 9: We updated the % penetrance in line 162 to match the previous citations.

Reviewer comment 10: Figure 2. SND and CBD are not cited in the text. Furthermore, the abbreviations are not defined.

Reply 10: We updated the text and included SND and CBD and defined them in the text.

It reads now, lines 211-213: “In iPD, approximately 5% of cases with clinically diagnosed PD present with post-mortem neuropathological findings of pure nigrostriatal degeneration or substantia nigra degeneration (SND), while 77% are LB disease including brainstem predominant, transitional, and diffuse LB disease.”

And lines 238-239: “Two cases with LRRK2 variants (p.R1707K and p.R2618P) also show neuropathology of cortico-basal degeneration (CBD).”

Reviewer comment 11: line 205.  R1441C/H are mentioned but not p.R1441G

Reply 11: Text is updated and reads now in line 220“…,which includes p.R1441C/G/H, p.Y1699C, p.G2019S, and p.I2020T.”

Reviewer comment 12: Table 2.TDP-43 is not present in any mutation (therefore, leftover). Most of additional differences are not such or links (are institutions or Universities).

Reply 12: We included a new reference for a TDP-43 case, which was suggested by the other reviewer. We deleted the column “additional references” with information about corresponding brainbanks that are not references and combined both columns.

 Reviewer comment 13: 2.2 Increased susceptibility to synthetic oxidants in LRRK2 animal models. It is not clear if this phenotype is exclusive of LRRK2 mutations, or PD-related mutations in general.

Reply 13: Important suggestion, we added the following text to the manuscript lines 324-328: “Similar to other dominant genetic PD transgenic models and models of gene-environment interactions on nigrostriatal toxicity, LRRK2 mutations seem to sensitize and augment the underlying toxicant-induced phenotype. Likewise, expression of SNCA p.A53T also increases susceptibility to toxicants. Conversely, overexpression of recessive PD gene products (modeling reversal of inherited mutation) is protective against toxicants. ”

Reviewer comment 14: 3.1 LRRK2 modifier risk variants: DNM3 also modifies AAO, but is not mentioned

Reply 14: DNM3 variant shows earlier age at onset with GG allele is mentioned in Table 3 and also added now to the text, line 341 “Candidate gene studies of LRRK2 modifiers for risk/penetrance and AAO have been described predominantly for p.G2019S, p.G2385R, p.R1628P, and other risk variants such as DNM3.”

Reviewer comment 15: l360-361 is in bold letters

Reply 15: Formatting is corrected.

Reviewer comment 16: ET (essential tremor) abbreviation is not defined in the text

Reply 16: We defined ET at first mention in text and also in the abbreviations section.

Reviewer comment 17: Summary sections is seems a Discussion or Perspective section

Reply 17: We renamed the Summary section to “Summary and perspective”.